# Transfusion: Understanding Transfer Learning for Medical Imaging

**Maithra Raghu**[*]
Cornell University and Google Brain
maithrar@gmail.com

**Chiyuan Zhang**[*]
Google Brain
chiyuan@google.com

**Jon Kleinberg**[†]
Cornell University
kleinber@cs.cornell.edu

**Samy Bengio**[†]
Google Brain
bengio@google.com

## Abstract

Transfer learning from natural image datasets, particularly IMAGENET, using standard large models and corresponding pretrained weights has become a de-facto method for deep learning applications to medical imaging. However, there are fundamental differences in data sizes, features and task specifications between natural image classification and the target medical tasks, and there is little understanding of the effects of transfer. In this paper, we explore properties of transfer learning for medical imaging. A performance evaluation on two large scale medical imaging tasks shows that surprisingly, transfer offers little benefit to performance, and simple, lightweight models can perform comparably to IMAGENET architectures. Investigating the learned representations and features, we find that some of the differences from transfer learning are due to the over-parametrization of standard models rather than sophisticated feature reuse. We isolate where useful feature reuse occurs, and outline the implications for more efficient model exploration. We also explore feature independent benefits of transfer arising from weight scalings.

## 1 Introduction

With the growth of deep learning, transfer learning has become integral to many applications — especially in medical imaging, where the present standard is to take an existing architecture designed for natural image datasets such as IMAGENET, together with corresponding pretrained weights (e.g. ResNet [10], Inception [27]), and then fine-tune the model on the medical imaging data.

This basic formula has seen almost universal adoption across many different medical specialties. Two prominent lines of research have used this methodology for applications in radiology, training architectures like ResNet, DenseNet on chest x-rays [31, 24] and ophthalmology, training Inception-v3, ResNet on retinal fundus images [2, 9, 23, 4]. The research on ophthalmology has also culminated in FDA approval [28], and full clinical deployment [29]. Other applications include performing early detection of Alzheimer's Disease [5], identifying skin cancer from dermatologist level photographs [6], and even determining human embryo quality for IVF procedures [15].

Despite the immense popularity of transfer learning in medical imaging, there has been little work studying its precise effects, even as recent work on transfer learning in the *natural image* setting [11, 16, 20, 12, 7] has challenged many commonly held beliefs. For example in [11], it is shown that

---

[*]Equal Contribution.
[†]Equal Contribution.

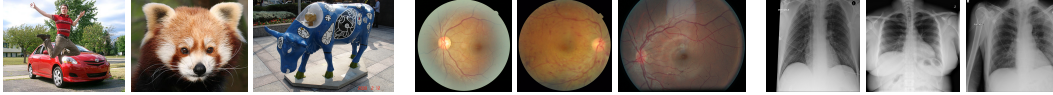

Figure 1: Example images from the IMAGENET, the *retinal fundus photographs*, and the CHEXPERT datasets, respectively. The fundus photographs and chest x-rays have much higher resolution than the IMAGENET images, and are classified by looking for small local variations in tissue.

transfer (even between similar tasks) does not necessarily result in performance improvements, while [16] illustrates that pretrained features may be less general than previously thought.

In the medical imaging setting, many such open questions remain. As described above, transfer learning is typically performed by taking a standard IMAGENET architecture along with its pretrained weights, and then fine-tuning on the target task. However, IMAGENET classification and medical image diagnosis have considerable differences.

First, many medical imaging tasks start with a large image of a bodily region of interest and use variations in local textures to identify pathologies. For example, in retinal fundus images, small red 'dots' are an indication of microaneurysms and diabetic retinopathy [1], and in chest x-rays local white opaque patches are signs of consolidation and pneumonia. This is in contrast to natural image datasets like IMAGENET, where there is often a clear global subject of the image (Fig. 1). There is thus an open question of how much IMAGENET feature reuse is helpful for medical images.

Additionally, most datasets have larger images (to facilitate the search for local variations), but with many fewer images than IMAGENET, which has roughly one million images. By contrast medical datasets range from several thousand images [15] to a couple hundred thousand [9, 24].

Finally, medical tasks often have significantly fewer classes (5 classes for Diabetic Retinopathy diagnosis [9], $5 - 14$ chest pathologies from x-rays [24]) than the standard IMAGENET classification setup of 1000 classes. As standard IMAGENET architectures have a large number of parameters concentrated at the higher layers for precisely this reason, the design of these models is likely to be suboptimal for the medical setting.

In this paper, we perform a fine-grained study on transfer learning for medical images. Our main contributions are:

**[1]** We evaluate the performance of standard architectures for natural images such as IMAGENET, as well as a family of non-standard but smaller and simpler models, on two large scale medical imaging tasks, for which transfer learning is currently the norm. We find that (i) in all of these cases, transfer does not significantly help performance (ii) smaller, simpler convolutional architectures perform comparably to standard IMAGENET models (iii) IMAGENET performance is not predictive of medical performance. These conclusions also hold in the very small data regime.

**[2]** Given the comparable performance, we investigate whether using pretrained weights leads to different learned representations, by using (SV)CCA [22] to directly analyze the hidden representations. We find that pretraining does affect the hidden representations, but there is a confounding issue of model size, where the large, standard IMAGENET models do not change significantly through the fine-tuning process, as evidenced through surprising correlations between representational similarity at initialization and after convergence.

**[3]** Using further analysis and weight transfusion experiments, where we partially reuse pretrained weights, we isolate locations where meaningful feature reuse does occur, and explore hybrid approaches to transfer learning where a subset of pretrained weights are used, and other parts of the network are redesigned and made more lightweight.

**[4]** We show there are also *feature-independent* benefits to pretraining — reusing only the *scaling* of the pretrained weights but not the features can itself lead to large gains in convergence speed.

## 2 Datasets

Our primary dataset, the RETINA data, consists of retinal *fundus photographs* [9], large $587 \times 587$ images of the back of the eye. These images are used to diagnose a variety of eye diseases including Diabetic Retinopathy (DR) [3]. DR is graded on a five-class scale of increasing severity [1]. Grades

3 and up are *referable DR* (requiring immediate specialist attention), while grades 1 and 2 correspond to *non-referable DR*. As in prior work [9, 2] we evaluate via AUC-ROC on identifying referable DR.

We also study a second medical imaging dataset, CHEXPERT [14], which consists of chest x-ray images (resized to $224 \times 224$), which can be used to diagnose 5 different thoracic pathologies: atelectasis, cardiomegaly, consolidation, edema and pleural effusion. We evaluate our models on the AUC of diagnosing each of these pathologies. Figure 1 shows some example images from both datasets and IMAGENET, demonstrating drastic differences in visual features among those datasets.

## 3 Models and Performance Evaluation of Transfer Learning

To lay the groundwork for our study, we select multiple neural network architectures and evaluate their performance when (1) training from random initialization and (2) doing transfer learning from IMAGENET. We train both standard, high performing IMAGENET architectures that have been popular for transfer learning, as well as a family of significantly smaller convolutional neural networks, which achieve comparable performance on the medical tasks.

As far as we are aware, there has been little work studying the effects of transfer learning from IMAGENET on smaller, non-standard IMAGENET architectures. (For example, [21] studies a different model, but does not evaluate the effect of transfer learning.) This line of investigation is especially important in the medical setting, where large, computationally expensive models might significantly impede mobile and on-device applications. Furthermore, in standard IMAGENET models, most of the parameters are concentrated at the top, to perform the 1000-class classification. However, medical diagnosis often has considerably fewer classes – both the retinal fundus images and chest x-rays have just 5 classes – likely meaning that IMAGENET models are highly overparametrized.

We find that across both datasets and all models, transfer learning does not significantly affect performance. Additionally, the family of smaller lightweight convolutional networks performs comparably to standard IMAGENET models, despite having significantly worse accuracy on IMAGENET– the IMAGENET task is not necessarily a good indication of success on medical datasets. Finally, we observe that these conclusions also hold in the setting of very limited data.

### 3.1 Description of Models

For the standard IMAGENET architectures, we evaluate ResNet50 [11] and Inception-v3 [27], which have both been used extensively in medical transfer learning applications [2, 9, 31]. We also design a family of simple, smaller convolutional architectures. The basic building block for this family is the popular sequence of a (2d) convolution, followed by batch normalization [13] and a relu activation. Each architecture has four to five repetitions of this basic layer. We call this model family CBR. Depending on the choice of the convolutional filter size (fixed for the entire architecture), the number of channels and layers, we get a family of architectures with size ranging from a third of the standard IMAGENET model size (CBR-LargeT, CBR-LargeW) to one twentieth the size (CBR-Tiny). Full architecture details are in the Appendix.

### 3.2 Results

We evaluate three repetitions of the different models and initializations (random initialization vs pretrained weights) on the two medical tasks, with the result shown in Tables 1, 2. There are two possibilities for repetitions of transfer learning: we can have a fixed set of pretrained weights and multiple training runs from that initialization, or for each repetition, first train from scratch on IMAGENET and then fine-tune on the medical task. We opt for evaluating the former, as that is the standard method used in practice. For all models except for Inceptionv3, we first train on IMAGENET to get the pretrained weights. For Inceptionv3, we used the pretrained weights provided by [26].

Table 1 shows the model performances on the RETINA data (AUC of identifying moderate Diabetic Retinopathy (DR), described in Section 2), along with IMAGENET top 5 accuracy. Firstly, we see that transfer learning has minimal effect on performance, not helping the smaller CBR architectures at all, and only providing a fraction of a percent gain for Resnet and Inception. Next, we see that despite the significantly lower performance of the CBR architectures on IMAGENET, they perform very comparably to Resnet and Inception on the RETINA task. These same conclusions are seen on

| Dataset | Model Architecture | Random Init | Transfer | Parameters | IMAGENET Top5 |
|---------|-------------------|-------------|----------|------------|----------------|
| RETINA | Resnet-50 | 96.4% ± 0.05 | 96.7% ± 0.04 | 23570408 | 92.% ± 0.06 |
| RETINA | Inception-v3 | 96.6% ± 0.13 | 96.7% ± 0.05 | 22881424 | 93.9% |
| RETINA | CBR-LargeT | 96.2% ± 0.04 | 96.2% ± 0.04 | 8532480 | 77.5% ± 0.03 |
| RETINA | CBR-LargeW | 95.8% ± 0.04 | 95.8% ± 0.05 | 8432128 | 75.1% ± 0.3 |
| RETINA | CBR-Small | 95.7% ± 0.04 | 95.8% ± 0.01 | 2108672 | 67.6% ± 0.3 |
| RETINA | CBR-Tiny | 95.8% ± 0.03 | 95.8% ± 0.01 | 1076480 | 73.5% ± 0.05 |

Table 1: **Transfer learning and random initialization perform comparably across both standard IMA-GENET architectures and simple, lightweight CNNs for AUCs from diagnosing moderate DR. Both sets of models also have similar AUCs, despite significant differences in size and complexity.** Model performance on DR diagnosis is also not closely correlated with IMAGENET performance, with the small models performing poorly on IMAGENET but very comparably on the medical task.

| Model Architecture | Atelectasis | Cardiomegaly | Consolidation | Edema | Pleural Effusion |
|--------------------|-------------|--------------|---------------|-------|-------------------|
| Resnet-50 | 79.52±0.31 | 75.23±0.35 | 85.49±1.32 | 88.34±1.17 | 88.70±0.13 |
| Resnet-50 (trans) | 79.76±0.47 | 74.93±1.41 | 84.42±0.65 | 88.89±1.66 | 88.07±1.23 |
| CBR-LargeT | 81.52±0.25 | 74.83±1.66 | 88.12±0.25 | 87.97±1.40 | 88.37±0.01 |
| CBR-LargeT (trans) | 80.89±1.68 | 76.84±0.87 | 86.15±0.71 | 89.03±0.74 | 88.44±0.84 |
| CBR-LargeW | 79.79±0.79 | 74.63±0.69 | 86.71±1.45 | 84.80±0.77 | 86.53±0.54 |
| CBR-LargeW (trans) | 80.70±0.31 | 77.23±0.84 | 86.87±0.33 | 89.57±0.34 | 87.29±0.69 |
| CBR-Small | 80.43±0.72 | 74.36±1.06 | 88.07±0.60 | 86.20±1.35 | 86.14±1.78 |
| CBR-Small (trans) | 80.18±0.85 | 75.24±1.43 | 86.48±1.13 | 89.09±1.04 | 87.88±1.01 |
| CBR-Tiny | 80.81±0.55 | 75.17±0.73 | 85.31±0.82 | 84.87±1.13 | 85.56±0.89 |
| CBR-Tiny (trans) | 80.02±1.06 | 75.74±0.71 | 84.28±0.82 | 89.81±1.08 | 87.69±0.75 |

Table 2: **Transfer learning provides mixed performance gains on chest x-rays.** Performances (AUC%) of diagnosing different pathologies on the CHEXPERT dataset. Again we see that transfer learning does not help significantly, and much smaller models performing comparably.

the chest x-ray results, Table 2. Here we show the performance AUC for the five different pathologies (Section 2). We again observe mixed gains from transfer learning. For Atelectasis, Cardiomegaly and Consolidation, transfer learning performs slightly worse, but helps with Edema and Pleural Effusion.

### 3.3 The Very Small Data Regime

We conducted additional experiments to study the effect of transfer learning in the very small data regime. Most medical datasets are significantly smaller than IMAGENET, which is also the case for our two datasets. However, our datasets still have around two hundred thousand examples, and other settings many only have a few thousand. To study the effects in this very small data regime, we trained models on only 5000 datapoints on the RETINA dataset, and examined the effect of transfer learning. The results, in Table 3, suggest that while transfer learning has a bigger effect with very

| Model | Rand Init | Pretrained |
|-------|-----------|------------|
| Resnet50 | 92.2% | 94.6% |
| CBR-LargeT | 93.6% | 93.9% |
| CBR-LargeW | 93.6% | 93.7% |

Table 3: **Benefits of transfer learning in the small data regime are largely due to architecture size.** AUCs when training on the RETINA task with only 5000 datapoints. We see a bigger gap between random initialization and transfer learning for Resnet (a large model), but not for the smaller CBR models.

small amounts of data, there is a confounding effect of model size – transfer primarily helps the large models (which are designed to be trained with a million examples) and smaller models again show little difference between transfer and random initialization.

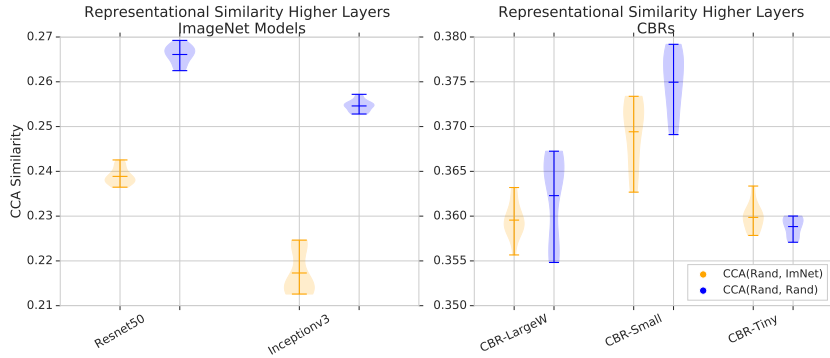

Figure 2: **Pretrained weights give rise to different hidden representations than training from random initialization for large models.** We compute CCA similarity scores between representations learned using pretrained weights and those from random initialization. We do this for the top two layers (or stages for Resnet, Inception) and average the scores, plotting the results in orange. In blue is a baseline similarity score, for representations trained from different random initializations. We see that representations learned from random initialization are more similar to each other than those learned from pretrained weights for larger models, with less of a distinction for smaller models.

## 4 Representational Analysis of the Effects of Transfer

In Section 3 we saw that transfer learning and training from random initialization result in very similar performance across different neural architectures and tasks. This gives rise to some natural questions about the effect of transfer learning on the kinds of *representations* learned by the neural networks. Most fundamentally, does transfer learning in fact result in any representational differences compared to training from random initialization? Or are the effects of the initialization lost? Does feature reuse take place, and if so, where exactly? In this section, we provide some answers to these basic questions. Our approach directly analyzes and compares the hidden representations learned by different populations of neural networks, using (SV)CCA [22, 19], revealing an important dependence on *model size*, and differences in behavior between lower and higher layers. These insights, combined with results Section 5 suggest new, hybrid approaches to transfer learning.

**Quantitatively Studying Hidden Representations with (SV)CCA** To understand how pretraining affects the features and representations learned by the models, we would like to (quantitatively) study the learned intermediate functions (latent layers). Analyzing latent representations is challenging due to their complexity and the lack of any simple mapping to inputs, outputs or other layers. A recent tool that effectively overcomes these challenges is (Singular Vector) Canonical Correlation Analysis, (SV)CCA [22, 19], which has been used to study latent representations through training, across different models, alternate training objectives, and other properties [22, 19, 25, 18, 8, 17, 30]. Rather than working directly with the model parameters or neurons, CCA works with *neuron activation vectors* — the ordered collection of outputs of the neuron on a sequence of inputs. Given the activation vectors for two sets of neurons (say, corresponding to distinct layers), CCA seeks linear combinations of each that are as correlated as possible. We adapt existing CCA methods to prevent the size of the activation sets from overwhelming the computation in large models (details in Appendix C), and apply them to compare the latent representations of corresponding hidden layers of different pairs of neural networks, giving a CCA similarity score of the learned intermediate functions.

**Transfer Learning and Random Initialization Learn Different Representations** Our first experiment uses CCA to compare the similarity of the hidden representations learned when training from pretrained weights to those learned when training from random initialization. We use the representations learned at the top two layers (for CBRs) or stages (for Resnet, Inception) before the output layer, averaging their similarity scores. As a baseline to compare to, we also look at CCA similarity scores for the same representations when training from random initialization with two different seeds (different initializations and gradient updates). The results are shown in Figure 2. For larger models (Resnet, Inception), there is a clear difference between representations, with the similarity of representations between training from random initialization and pretrained weights (orange) noticeably lower

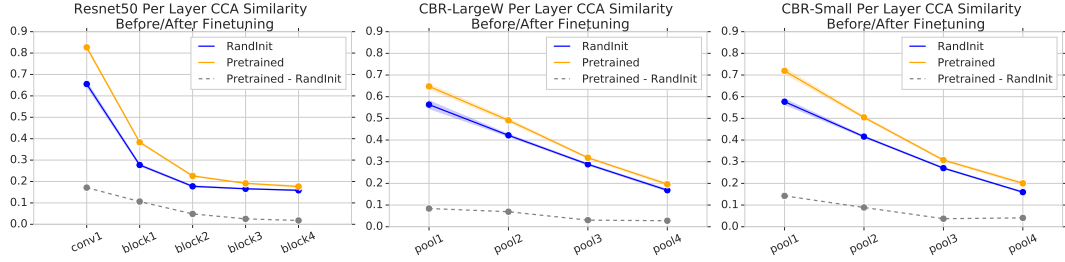

Figure 3: **Per-layer CCA similarities before and after training on medical task.** For all models, we see that the lowest layers are most similar to their initializations, and this is especially evident for Resnet50 (a large model). We also see that feature reuse is mostly restricted to the bottom two layers (stages for Resnet) — the only place where similarity with initialization is significantly higher for pretrained weights (grey dotted lines shows the difference in similarity scores between pretrained and random initialization).

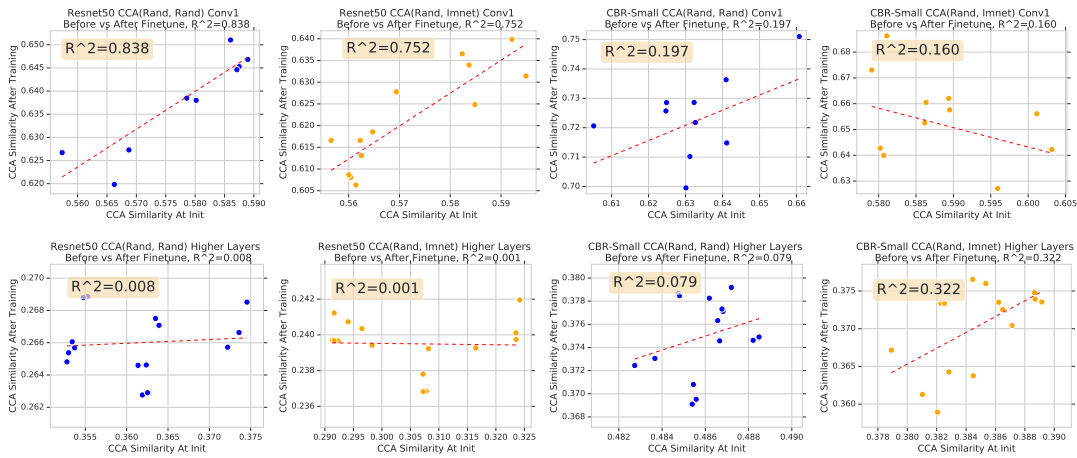

Figure 4: **Large models move less through training at lower layers: similarity at initialization is highly correlated with similarity at convergence for large models.** We plot CCA similarity of Resnet (conv1) initialized randomly and with pretrained weights at (i) initialization, against (ii) CCA similarity of the converged representations (top row second from left.) We also do this for two different random initializations (top row, left). In *both* cases (even for random initialization), we see a surprising, strong correlation between similarity at initialization and similarity after convergence ($R^2 = 0.75, 0.84$). This is not the case for the smaller CBR-Small model, illustrating the overparametrization of Resnet for the task. Higher must likely change much more for good task performance.

than representations learned independently from different random initializations (blue). However for smaller models (CBRs), the functions learned are more similar.

**Larger Models Change Less Through Training** The reasons underlying this difference between larger and smaller models becomes apparent as we further study the hidden representations of all the layers. We find that *larger models change much less during training*, especially in the lowest layers. This is true *even when they are randomly initialized*, ruling out feature reuse as the sole cause, and implying their overparametrization for the task. This is in line with other recent findings [33].

In Figure 3, we look at per-layer representational similarity before/after finetuning, which shows that the lowest layer in Resnet (a large model), is significantly more similar to its initialization than in the smaller models. This plot also suggests that any serious feature reuse is restricted to the lowest couple of layers, which is where similarity before/after training is clearly higher for pretrained weights vs random initialization. In Figure 4, we plot the CCA similarity scores between representations using pretrained weights and random initialization *at initialization* vs after training, for the lowest layer (conv1) as well as higher layers, for Resnet and CBR-Small. Large models changing less through training is evidenced by a surprising correlation between the CCA similarities for Resnet conv1, which is not true for higher layers or the smaller CBR-Small model.

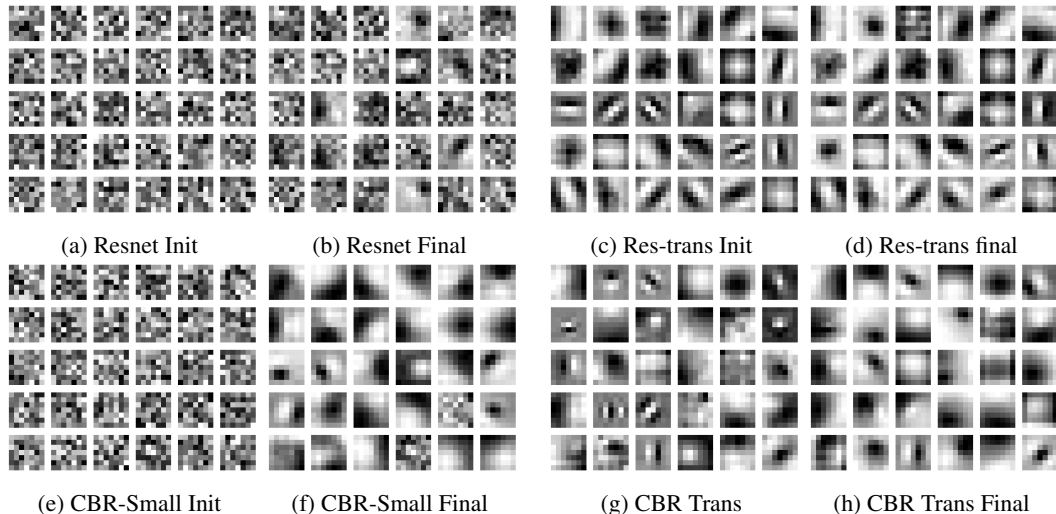

|  |  |  |  |
|---|---|---|---|
| (a) Resnet Init | (b) Resnet Final | (c) Res-trans Init | (d) Res-trans final |
| (e) CBR-Small Init | (f) CBR-Small Final | (g) CBR Trans | (h) CBR Trans Final |

Figure 5: **Visualization of conv1 filters shows the remains of initialization after training in Resnet, and the lack of and erasing of Gabor filters in CBR-Small.** We visualize the filters before and after training from random initialization and pretrained weights for Resnet (top row) and CBR-Small (bottom row). Comparing the similarity of (e) to (f) and (g) to (h) shows the limited movement of Resnet through training, while CBR-Small changes much more. We see that CBR does not learn Gabor filters when trained from scratch (f), and also erases some of the pretrained Gabors (compare (g) to (h).)

**Filter Visualizations and the Absence of Gabors** As a final study of how pretraining affects the model representations, we visualize some of the filters of conv1 for Resnet and CBR-Small (both 7x7 kernels), before and after training on the RETINA task. The filters are shown in Figure 5, with visualizations for chest x-rays in the Appendix. These add evidence to the aforementioned observation: the Resnet filters change much less than those of CBR-Small. In contrast, CBR-Small moves more from its initialization, and has more similar learned filters in random and pretrained initialization. Interestingly, CBR-Small does not appear to learn Gabor filters when trained from scratch (bottom row second column). Comparing the third and fourth columns of the bottom row, we see that CBR-Small even erases some of the Gabor filters that it is initialized with in the pretrained weights.

## 5 Convergence: Feature Independent Benefits and Weight Transfusion

In this section, we investigate the effects of transfer learning on convergence speed, finding that: (i) surprisingly, transfer offers *feature independent* benefits to convergence simply through better weight scaling (ii) using pretrained weights from the lowest two layers/stages has the biggest effect on convergence — further supporting the findings in the previous section that any meaningful feature reuse is concentrated in these lowest two layers (Figure 3.) These results suggest some hybrid approaches to transfer learning, where only a subset of the pretrained weights (lowest layers) are used, with a lightweight redesign to the top of the network, and even using entirely *synthetic* features, such as synthetic Gabor filters (Appendix F.3). We show these hybrid approaches capture most of the benefits of transfer and enable greater flexibility in its application.

**Feature Independent Benefits of Transfer: Weight Scalings** We consistently observe that using pretrained weights results in faster convergence. One explanation for this speedup is that there is significant feature reuse. However, the results of Section 4 illustrate that there are many confounding factors, such as model size, and feature reuse is likely limited to the lowest layers. We thus tested to see whether there were *feature independent* benefits of the pretrained weights, such as *better scaling*. In particular, we initialized a *iid weights* from $\mathcal{N}(\tilde{\mu}, \tilde{\sigma}^2)$, where $\tilde{\mu}$ and $\tilde{\sigma}^2$ are the mean and variance of $\tilde{W}$, the pretrained weights. Doing this for each layer separately inherits the scaling of the pretrained weights, but destroys all of the features. We called this the *Mean Var* init, and found that it significantly helps speed up convergence (Figure 6.) Several additional experiments studying batch normalization, weight sampling, etc are in the Appendix.

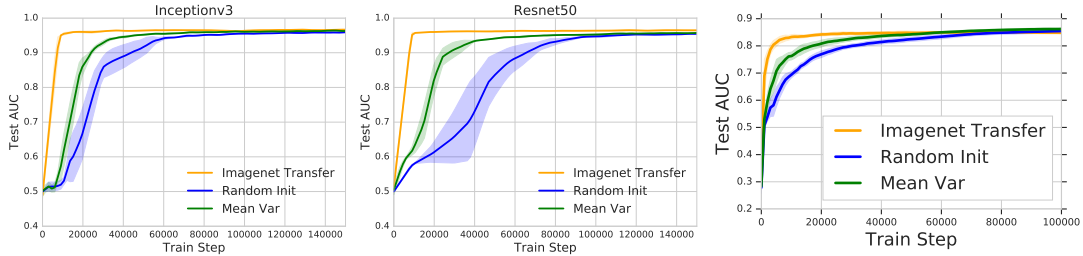

Figure 6: **Using only the scaling of the pretrained weights (Mean Var Init) helps with convergence speed**. The figures compare the standard transfer learning and the *Mean Var* initialization scheme to training from scratch. On both the RETINA data (a-b) and the CHEXPERT data (c) (with Resnet50 on the *Consolidation* disease), we see convergence speedups.

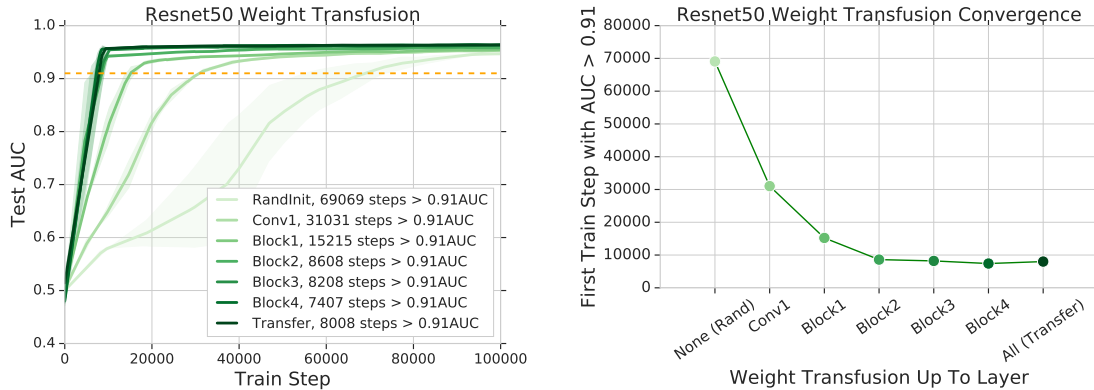

Figure 7: **Reusing a subset of the pretrained weights (weight transfusion), further supports only the lowest couple of layers performing meaningful feature reuse.** We initialize a Resnet with a contiguous subset of the layers using pretrained weights (weight transfusion), and the rest randomly, and train on the RETINA task. On the left, we show the convergence plots when transfusing up to conv1 (just one layer), up to block 1 (conv1 and all the layers in block1), etc up to full transfer. On the right, we plot the number of train steps taken to reach $91\%$ AUC for different numbers of transfused weights. Consistent with findings in Section 4, we observe that reusing the lowest layers leads to the greatest gain in convergence speed. Perhaps surprisingly, just reusing conv1 gives the greatest marginal convergence speedup, even though transfusing weights for a block means several new layers are using pretrained weights.

**Weight Transfusions and Feature Reuse** We next study whether the results suggested by Section 4, that meaningful feature reuse is restricted to the lowest two layers/stages of the network is supported by the effect on convergence speed. We do this via a *weight transfusion* experiment, transfering a contiguous set of some of the pretrained weights, randomly initializing the rest of the network, and training on the medical task. Plotting the training curves and steps taken to reach a threshold AUC in Figure 7 indeed shows that using pretrained weights for lowest few layers has the biggest training speedup. Interestingly, just using pretrained weights for conv1 for Resnet results in the largest gain, despite transfusion for a Resnet block meaning multiple layers are now reusing pretrained weights.

**Takeaways: Hybrid Approaches to Transfer Learning** The transfusion results suggest some hybrid, more flexible approaches to transfer learning. Firstly, for larger models such as Resnet, we could consider reusing pretrained weights up to e.g. Block2, redesiging the top of the network (which has the bulk of the parameters) to be more lightweight, initializing these layers randomly, and training this new Slim model end to end. Seeing the disproportionate importance of conv1, we might also look at the effect of initializing conv1 with *synthetic Gabor filters* (see Appendix F.3 for details) and the rest of the network randomly. In Figure 8 we illustrate these hybrid approaches. Slimming the top of the network in this way offers the same convergence and performance as transfer learning, and using synthetic Gabors for conv1 has *the same effect* as pretrained weights for conv1. These variants highlight many new, rich and flexible ways to use transfer learning.

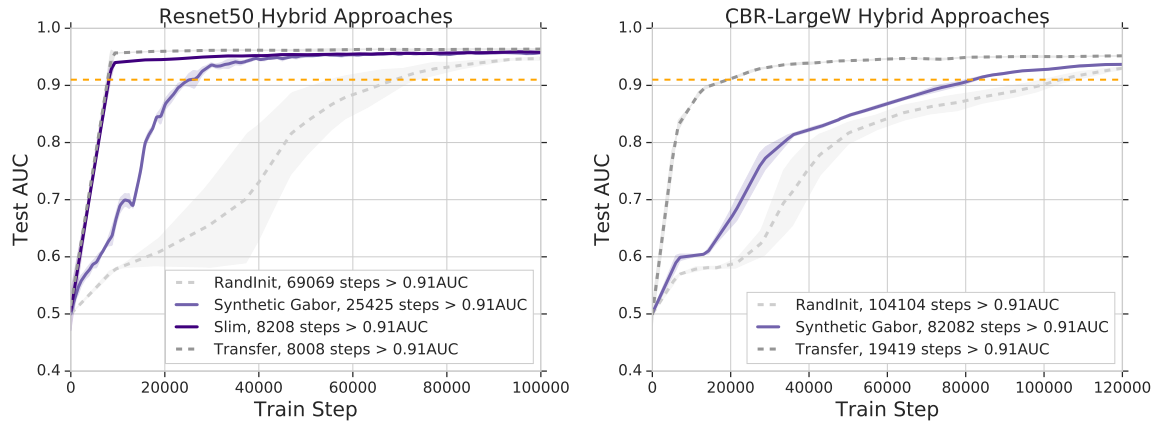

Figure 8: **Hybrid approaches to transfer learning: reusing a subset of the weights and slimming the remainder of the network, and using synthetic Gabors for conv1.** For Resnet, we look at the effect of reusing pretrained weights up to Block2, and slimming the remainder of the network (halving the number of channels), randomly initializing those layers, and training end to end. This matches performance and convergence of full transfer learning. We also look at initializing conv1 with *synthetic Gabor filters* (so *no* use of pretrained weights), and the rest of the network randomly, which performs equivalently to reusing conv1 pretrained weights. This result generalizes to different architectures, e.g. CBR-LargeW on the right.

# 6 Conclusion

In this paper, we have investigated many central questions on transfer learning for medical imaging applications. Having benchmarked both standard IMAGENET architectures and non-standard lightweight models (itself an underexplored question) on two large scale medical tasks, we find that transfer learning offers limited performance gains and much smaller architectures can perform comparably to the standard IMAGENET models. Our exploration of representational similarity and feature reuse reveals surprising correlations between similarities at initialization and after training for standard IMAGENET models, providing evidence of their overparametrization for the task. We also find that meaningful feature reuse is concentrated at the lowest layers and explore more flexible, hybrid approaches to transfer suggested by these results, finding that such approaches maintain all the benefits of transfer and open up rich new possibilities. We also demonstrate feature-independent benefits of transfer learning for better weight scaling and convergence speedups.

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
