[Supplementary Material · Transfusion_Paper-12-22.pdf]

# Appendix to "Transfusion: Understanding Transfer Learning for Medical Imaging"

## A   Details on Datasets, Models and Hyperparameters

The RETINA dataset consisted of around 250k training images, and 70k test images. The train test split was done by patient id (as is standard for medical datasets) to ensure no accidental similarity between the train/test dataset. The chest x-ray dataset is open sourced and available from [14], which has all of the details. Briefly, they have 223k training images and binary indicator for multiple diseases assiciated with each image extracted automatically from the meta data. The standard IMAGENET (ILSVRC 2012) dataset was also used to pretrain models.

For dataset preprocessing we used mild random cropping, as well as standard normalization by the mean and standard deviation for IMAGENET. We augmented the data with hue and contrast augmentations. For the RETINA data, we used random horizontal and vertical flips, and for the chest x-ray data, we did not do random flip. We did not do model specific hyperparameter tuning on each target data, and used fixed standard hyperparameters.

For experiments on the RETINA data, we trained the standard IMAGENET models, Resnet50 and Inception-v3, by replacing the final 1000 class IMAGENET classification head with a five class head for DR diagnosis, or five classes for the five different chest x-ray diseases. We use the sigmoid activation at the top instead of the multiclass softmax activation, and the train the models in the multi-label binary classification framework.

The CBR family of small convolutional neural networks consists of multiple conv2d-batchnorm-relu layers followed by a maxpool. Each maxpool has spatial window (3x3) and stride (2x2). For each CBR architecture, there is one filter size for all the convolutions (which all have stride 1). Below, conv-n denotes a 2d convolutionl with n output channels.

- **CBR-LargeT**(all) has 7x7 conv filters: (conv32-bn-relu) maxpool (conv64-bn-relu) maxpool (conv128-bn-relu) maxpool (conv256-bn-relu) maxpool (conv512-bn-relu) global avgpool, classification
- **CBR-LargeW**(ide) has 7x7 conv filters: (conv64-bn-relu) maxpool (conv128-bn-relu) maxpool (conv256-bn-relu) maxpool (conv512-bn-relu) maxpool, global avgpool, classification.
- **CBR-Small** has 7x7 conv filters: (conv32-bn-relu) maxpool (conv64-bn-relu) maxpool (conv128-bn-relu) maxpool (conv256-bn-relu) maxpool global avgpool, classification
- **CBR-Tiny** has 5x5 conv filters: (conv64-bn-relu) maxpool (conv128-bn-relu) maxpool (conv256-bn-relu) maxpool (conv512-bn-relu) maxpool, global avgpool, classification.

The models on RETINA are trained on $587 \times 587$ images, with learning rate 0.001 and a batch size of 8 (for memory considerations.) The Adam optimizer is used. The models on the chest x-ray are trained on $224 \times 224$ images, with a batch size of 32, and vanilla SGD with momentum (coefficient 0.9). The learning rate scheduling is inherited from the IMAGENET training pipeline, which warms up from 0 to $0.1 \times \frac{32}{256}$ in 5 epochs, and then decay with a factor of 10 on epoch 30, 60, and 90, respectively.

## B   Additional Dataset Size Results

Complementing the data varying experiments in the main text, we additional experiments on varying the amount of training data, fidning that for around 50k datapoints, we return to only seeing a fractional improvement of transfer learning. Future work could study how hybrid approaches perform when less data is available.

## C   CCA Details

For full details on the implementation of CCA, we reference prior work [22, 19], as well as the open sourced code (the source of our implementation): https://github.com/google/svcca

| Model | Init Method | 5k | 10k | 50k | 100k |
|-------|-------------|-----|-----|-----|------|
| Resnet50 | IMAGENET Pretrained | 94.6% | 94.8% | 95.7% | 96.0 |
| Resnet50 | Random Init | 92.2% | 93.3% | 95.3% | 95.9% |
| CBR-LargeT | Random Init | 93.6% | - | - | - |
| CBR-LargeT | Pretrained | 93.9% | - | - | - |
| CBR-LargeW | Random Init | 93.6% | - | - | - |
| CBR-LargeW | Pretrained | 93.7% | - | - | - |
| Resnet50 | Conv1 Pretrained | 92.9% | - | - | - |
| Resnet50 | Mean Var Init | - | 94.4% | 95.5% | 95.8% |

Table 4: **Additional performance results when varying initialization and the dataset size on the RETINA task.** For Resnet50, we show performances when training on very small amounts of data. We see that even finetuning (with early stopping) on 5k datapoints beats the results from performing fixed feature extraction, Figure 12, suggesting finetuning should always be preferred. For 5k, 10k datapoints, we see a larger gap between transfer learning and random init (closed by 50k datapoints) but this is likely due to the enormous size of the model (typically trained on 1 million datapoints) compared to the dataset size. This is supported by evaluating the effect of transfer on CBR-LargeT and CBR-LargeW, where transfer again does not help much. (These are one third the size of Resnet50, and we expect the gains of transfer to be even more minimal for CBR-Small and CBR-Tiny.) We also show results for using the MeanVar init, and see some gains in performance for the very small data setting. We also see a small gain on 5k datapoints when just reusing the conv1 weights for Resnet50.

One challenge we face when implementing CCA is the large size of the convolutional activations. These activations have shape $(n, h, w, c)$, where $n$ is the number of datapoints, $c$ the number of channels, and $h, w$ the spatial dimensions. These values all vary significantly across the network, e.g. conv1 has shape $(n, 294, 294, 64)$, while activations at the end of block 3 have shape $(n, 19, 19, 1024)$. Because CCA is sensitive to both the number of datapoints $n$ (actually $hwn$ for convolutional layers) and the number of neurons – $c$ for large convolutional layers – there is large variations in scaling across different layers in the model. To address this, we do the following: let $L$ and $L'$ be the layers we want to compare, with shape (height, width, channels), $(h_L, w_L, c_L)$. We apply CCA as follows:

- Pick $p$, the total number of image patches to compute activation vectors and CCA over, and $d$, the maximum number of neuron activation vectors to correlate with

- Pick the number of datapoints $n$ so that $nh_Lw_L = p$.

- Sample $d$ of the $c_L$ channels, and apply CCA to the resulting $d$ x $nh_Lw_L$ activation matrices.

- Repeat over samples of $d$ and $n$.

This works much better than prior approaches of averaging over all of the spatial dimensions [19], or flattening across all of the neurons [22] (too computationally expensive in this setting.)

## D    Additional Results from Representation Analysis

Here, we include some additional results studying the representations of these models. We perform more representational similarity comparisons between networks trained from (the same) pretrained weights (as is standard), but different random seeds. We do this for Resnet50 (a large model) and CBR-Small (a small model), and Table 5 includes these results as well as similarity comparisons for networks trained with different random seeds and *different* random initializations as a baseline. The comparisons across layers and models is slightly involved, but as we detail below, the evidence further supports the conclusions in the main text:

- *Larger models change less through training.* Comparing CCA similarity scores across models is a little challenging, due to different scalings, so we look at the difference in CCA similarity between two models trained with pretrained weights, and two models trained from random initialization, for Resnet50 and CBR-Small. Comparing this value for Conv1 (in Resnet50) to Pool1 (in CBR-Small), we see that pretraining results in much more similar representations compared to random initialization in the large model over the small model.

| Description | Conv1 | Block1 | Block2 | Block3 | Block4 |
|---|---|---|---|---|---|
| Resnet50 CCA(ImNet1, ImNet2) | 0.865 | 0.559 | 0.421 | 0.343 | 0.313 |
| Resnet50 CCA(Rand1, Rand2) | 0.647 | 0.369 | 0.277 | 0.256 | 0.276 |
| Resnet50 Diff | **0.218** | **0.191** | **0.144** | **0.086** | **0.037** |

| Description | Pool1 | Pool2 | Pool3 | Pool4 |
|---|---|---|---|---|
| CBR-Small CCA(ImNet1, ImNet2) | 0.825 | 0.709 | 0.477 | 0.395 |
| CBR-Small CCA(Rand1, Rand2) | 0.723 | 0.541 | 0.401 | 0.349 |
| CBR-Small Diff | **0.102** | **0.168** | **0.076** | **0.046** |

Table 5: **Representational comparisons between trained ImageNet models with different seeds highlight the variation of behavior in higher and lower layers, and differences between larger and smaller models.** We compute CCA similarity between representations at different layers when training from different random seeds with (i) (the same) pretrained weights (ii) different random inits, for Resnet and CBR-Small. The results support the conclusions of the main text. For Resnet50, in the lowest layers such as Conv1 and Block1, we see that representations learned when using (the same) pretrained weights are much more similar to each other (diff 0.2 in CCA score) than representations learned from different random initializations. This $\sim 0.2$ difference is also much higher than (somewhat) corresponding differences in CBR-Small, for Pool1, Pool2. Actually, as Resnet50 is much deeper, the large difference in Block1 is very striking. (Block 1 alone contains much more layers than all of CBR-Small.) By Block3 and Block4 however, the CCA similarity difference between pretrained representations and those from random initialization is much smaller, and slightly lower than the differences for Pool3, Pool4 in CBR-Small, suggesting that pretrained weights are not having much of a difference on the kinds of functions learned. For CBR-Small, we also see that pretrained weights result in larger differences between the representations in the lower layers, but these become much smaller in the higher layers. We also observe that representations in CBR-Small trained from random initialization (especially in the lower layers e.g. Pool1) are more similar to each other than in Resnet50, suggesting things move more.

    (a) rand init      (b) final (rand init)      (c) transfer init      (d) final (transfer init)

Figure 9: **First layer filters of CBR-Small on the CHEXPERT data.** (a) and (c) show the randomly initialized filters and filters initialized from a model (the same architecture) pre-trained on IMAGENET. (b) and (d) shows the final converged filters from the two different initializations, respectively.

- *The effect of pretraining is mostly limited to the lowest layers* For higher layers, the CCA similarities between representations using pretrained weights and those trained from random initializations are closer, and the difference between CBR-Small and Resnet-50 is non-existent, suggesting that the effects of pretraining mostly affect the lowest layers across models, with finetuning changing representations at the top.

Figure 9 and Figure 10 compare the first layer filters between transfer learning and training from random initialization on the CHEXPERT data for the CBR-Small and Resnet-50 architectures, respectively. Those results complement Figure 5 in the main text.

# E  The Fixed Feature Extraction Setting

To complete the picture, we also study the fixed feature extractor setting. While the most popular methodology for transfer learning is to initialize from pretrained weights and fine-tune (train) the entire network, an alternative is to initialize all layers up to layer $L$ with pretrained weights. These are then treated as a fixed feature extractor, with only layers $L + 1$ onwards, being trained. There are two variants of this fixed feature extractor experiment: **[1]** Initialize all layers with pretrained weights

| (a) rand init | (b) final (rand init) | (c) transfer init | (d) final (transfer init) |

Figure 10: **First layer filters of Resnet-50 on the CHEXPERT data.** (a) and (c) show the randomly initialized filters and filters initialized from a model (the same architecture) pre-trained on IMAGENET. (b) and (d) shows the final converged filters from the two different initializations, respectively.

Figure 11: **Larger models move less through training than smaller networks.** A schematic diagram of our intuition for optimization for larger and smaller models.

and only train layer $L + 1$ onwards. **[2]** Initialize only up to layer $L$ with pretrained weights, and layer $L + 1$ onwards randomly; then train only layers $L + 1$ onwards.

We implement both of these versions across different models trained on the RETINA task in Figure 12, and CHEXPERT in Figure 13, including a baseline of using random features – initializing the network randomly, freezing up to layer $L$, and training layer $L + 1$ onwards. For the RETINA task, we see that the pretrained IMAGENET features perform significantly better than the random features baseline, but this gap is significantly closer on the chest x-rays.

More surprisingly however, there is little difference in performance between initializing all layers with pretrained weights and only up to layer $L$ with pretrained weights. This latter experiment has also been studied in [32], where they found that re-initializing caused drops in performance due to *co-adaptation*, where neurons in different layers have evolved together in a way that is not easily discoverable through retraining. This analysis was done for highly similar tasks (different subsets of IMAGENET), and we hypothesise that in our setting, the significant changes of the higher layers (Figures 3, 4) means that the correct adaptation is naturally learned through training.

## F  Additional Results on Feature Independent Benefits and Weight Transfusions

Figure 14 visualizes the first layer filters from various initialization schemes. As shown in the main text, the *Mean Var* initialization could converge much faster than the baseline random initialization due to better parameter scaling transferred from the pre-trained weights. Figure 15 shows more

Figure 12: **IMAGENET features perform well as fixed feature extractors on the RETINA task, and are robust to coadaptation performance drops.** We initialize (i) the full architecture with IMAGENET weights (yellow) (ii) up to layer $L$ with IMAGENET weights, and the rest randomly. In both, we keep up to layer $L$ fixed, and only train layers $L + 1$ onwards. We compare to a random features baseline, initializing randomly and training layer $L + 1$ onwards (blue). IMAGENET features perform much better as fixed feature extractors than the random baseline (though this gap is much closer for the CHEXPERT dataset, Appendix Figure 13.) Interestingly, there is no performance drop due to the *coadaptation* issue [32], with partial IMAGENET initialization performing equally to initialzing with all of the IMAGENET weights.

results on RETINA with various architectures. We find that on smaller models, the effectiveness of the *Mean Var* initialization is less very pronounced, likely due to them being much shallower.

Figure 16 shows all the five diseases on the CHEXPERT data for Resnet-50. Except for Cardiomegaly, we see benefits of the *Mean Var* initialization scheme on convergence speed in all other diseases.

## F.1 Batch Normalization Layers

Batch normalization layers Ioffe and Szegedy [13] are an essential building block for most modern network architectures with visual inputs. However, these layers have a slightly different structure that requires more careful consideration when performing the Mean Var init. Letting $x$ be a batch of activations, batch norm computes

$$\gamma \left( \frac{(x - \mu_B)}{\sigma_B + \epsilon} \right) + \beta$$

Here, $\gamma, \beta$ are learnable scale, shift parameters, and $\mu_B, \sigma_B$ are an accumulated running mean and variance over the train dataset. Thus, in transfer learning, $\mu_B, \sigma_B$ start off as the mean/variance of the IMAGENET data activations, unlikely to match the medical image statistics. Therefore, for the Mean Var Init, we initialized all of the batch norm parameters to the identity: $\gamma, \sigma_B = 1$, $\beta, \mu_B = 0$. We call this the *BN Identity Init*. Two alternatives are *BN* IMAGENET *Mean Var*, resampling the values of all batch norm parameters according to the IMAGENET means and variances, and *BN* IMAGENET *Transfer*, copying over the batch norm parameters from IMAGENET. We compare these three methods in Figure 17, with non-batchnorm layers initialized according to the Mean Var Init. Broadly, they perform similarly, with *BN Identity Init* (used by default in other Mean Var related experiments) performing slightly better. We observe that *BN* IMAGENET *Transfer*, where the IMAGENET batchnorm parameters are transferred directly to the medical images, performs the worst.

## F.2 Mean Var Init vs Using Knowledge of the Full Empirical IMAGENET Weight Distribution

In Figure 14, we see that while the Mean Var Init might have the same mean and variance as the IMAGENET weight distribution, the two distributions themselves are quite different from each other. We examined the convergence speed of initializing with the Mean Var Init vs initializing using knowledge of the entire empirical distribution of the IMAGENET weights.

In particular, we looked at (1) *Sampling Init:* each weight is drawn iid from the full empirical distribution of IMAGENET weights (2) *Shuffled Init:* random shuffle of the pretrained IMAGENET weights to form a new initialization. (Note this is exactly sampling from the empirical distribution without replacement.) The results are illustrated in Figure 18. Interestingly, Mean Var is very similar

Figure 13: **Experiments on freezing lower layers of CBR-LargeT and a CBR-Tiny model on the CHEX-PERT data.** After random or transfer initialization, we keep up to layer $L$ fixed, and only train layers $L + 1$ onwards. IMAGENET features perform better as fixed feature extractors than the random baseline for most diseases, but the gap is much closer than for the RETINA data, Figure 12. We again see that there is no significant performance drop due to coadaptation challenges.

in convergence speed to both of these alternatives. This would suggest that further improvements in convergence speed might have to come from also modelling correlations between weights.

## F.3   Synthetic Gabor Filters

We test mathematically synthetic Gabor filters in place of learned Gabor filters on IMAGENET for its benefits in speeding up the convergence when used as initialization in the first layer of neural networks. The Gabor filters are generated with the skimage package, using the following code snippet.

```
from skimage.filters import gabor_kernel
from skimage.transform import resize
import numpy as np

def gen_gabors(n_angles=16, sigmas=[2], freqs = [0.08, 0.16, 0.25, 0.32],
               kernel_resize = 10, kernel_crop = 7):
  kernels = []
```

Figure 14: **Distribution and filter visualization of weights initialized according to pretrained IMAGENET weights, Random Init, and Mean Var Init**. The top row is a histogram of the weight values of the the first layer of the network (Conv 1) when initialized with these three different schemes. The bottom row shows some of the filters corresponding to the different initializations. Only the IMAGENET Init filters have pretrained (Gabor-like) structure, as Rand Init and Mean Var weights are iid.

Figure 15: **Comparison of convergence speed for different initialization schemes on RETINA with various model architectures.** The three plots present the results for CBR-LargeW, CBR-Small and CBR-Tiny, respectively.

```
for sigma in sigmas:
  for frequency in freqs:
    for theta in range(n_angles):
      theta = theta / n_angles * np.pi
      kernel = np.real(gabor_kernel(frequency, theta=theta,
                                    sigma_x=sigma, sigma_y=sigma))
      kernel_size = kernel.shape[0]
      if kernel_size > kernel_resize:
        kernel = resize(kernel, (kernel_resize, kernel_resize))
        kernel_size = kernel.shape[0]
      else:
        assert kernel_size >= kernel_crop
      # center crop
      size_delta = kernel_size - kernel_crop
      kernel = kernel[size_delta//2:-(size_delta-size_delta//2),
                      size_delta//2:-(size_delta-size_delta//2)]

      kernels.append(kernel)
```

(a) Atelectasis                (b) Cardiomegaly                (c) Consolidation

(d) Edema                                    (e) Pleural Effusion

Figure 16: **Comparison of convergence speed for different initialization schemes on the CHEX-PERT data with Resnet-50.**

Figure 17: **Comparing different ways of importing the weights and statistics for batch normalization layers.** The rest of the layers are initialized according to the Mean Var scheme. The two dashed lines show the convergence of the IMAGENET init and the Random init for references. The lines are averaged over 5 runs.

Figure 18: **The Mean Var Init converges with a similar speed to using the full empirical distribution of the pretrained IMAGENET weights.** The plots show the convergence speed of initializing by sampling from the empirical IMAGENET weight distribution, and initializing by randomly shuffling the pretrained weights (i.e. sampling without replacement). We see that Mean Var converges at a similar speed to using the full empirical distribution. All lines are averaged over 3 runs, and the dashed lines show the convergence of the IMAGENET init and the Random init as a reference.

Figure 19: **Weight transfusion results on Resnet50 (from main text) and CBR-LargeW**. These broadly show the same results — reusing pretrained weights for lowest layers give significantly larger speedups. Because CBR-LargeW is a much smaller model, there is slightly more change when reusing pretrained weights in high layers, but we still see the same diminishing returns pattern.

Figure 20: **Convergence of Slim Resnet50 from random initialization**. We include the convergence of the slim Resnet50 — where layers in Block3, Block4 have half the number of channels, and when we don't use any pretrained weights. We see that it is significantly slower than the hybrid approach in the main text.

Figure 21: Synthetic Gabor filters used to initialize the first layer if neural networks in some of the experiments in this paper. The Gabor filters are generated as grayscale images and repeated across the RGB channels.

```
    return kernels
```

Figure 21 visualize the synthetic Gabor filters. To ensure fair comparison, the synthesized Gabor filters are scaled (globally across all the filters with a single scale parameter) to match the numerical magnitudes of the learned Gabor filters.