[Reviews · NeurIPS 2019]

Reviewer 1



The authors investigate the current transfer learning scheme for deep learning applications to medical imaging. They thoroughly assess and compare the performance of standard architectures that originally designed for the natural image classification tasks with their in-house-developed lightweight and simple models on medical imaging tasks. In this concern, the study demonstrates that latter models can perform comparably with computationally expensive state-of-the-art models. The second finding of the study is that transfer learning does not have a significant benefit for performance. The authors validate the claim by comparing the latent representations of the networks learned with the pretrained weights and training from scratch, and by measuring representational similarity with canonical correlation analysis (CCA). However, in Figure 2, the authors mention that networks from random initialization are more similar representationally to each other. The phenomenon is true for standard pretrained models and smaller versions (CBR-Small and CBR-Tiny) of the proposed models. In contrast, we notice little inconsistencies/mixed performances for two other proposed models (CBR-LargeT and CBR-LargeW). It would be nice if the authors could analyze the situation in the manuscript. The paper illustrates the key properties of transfer learning for medical imaging, analyzing how transfer learning and pretrained weight initialization affect the features and representations learned by the network. The study evidently demonstrates why complex models and pretrained weights are not essential for medical imaging. Also, the work presents feature-independent benefits of transfer learning for better weight scaling and convergence speedups. The motivation of the work is clearly mentioned and explained, and the organization of the study is well-structured. The findings of the study will influence the researchers to develop lightweight deep learning models for the medical imaging domain in upcoming days. However, I am concerned about the originality of the work and significance of the study, considering a high bar to NeurIPS. First of all, the researchers already showed that a simpler network could preserve comparable performance with state-of-the-art networks, and one such study is tuberculosis diagnosis with a CNN [1]. I think this is an incremental contribution to existing work with detailed experiments. Secondly, the study is limited to classification tasks; however, it would be nice if the authors could provide insightful analysis for other medical tasks, such as segmentation, localization, quantification, and so on. Overall, I appreciate the authors’ efforts for their work. One of the main strengths of the study is that with the detailed findings of the paper, the researchers may focus on designing lightweight models for deep learning to medical imaging tasks. Since heavier models significantly impede mobile and on-device applications, the medical domain demands a simpler model in reality. With the limited, but for an overall important contribution, I think the paper marginally above the acceptance threshold. Hence, I vote for accepting the paper. [1] F. Pasa, V. Golkov, F. Pfeiffer, D. Cremers & D. Pfeiffer, ‘Efficient Deep Network Architectures for Fast Chest X-Ray Tuberculosis Screening and Visualization’. Scientific reports, 9(1):6268, 2019. Response to rebuttal: The authors have addressed some of our concerns in their feedback. They have tried to support their hypothesis whether the deep networks and the transfer learning are useful or not in the medical domain through this study. In fact, these are the central questions for many other domains including healthcare, and I have mentioned one of the studies, which shows that simpler network could preserve comparable performance with state-of-the-art networks [1]. However, the authors have not dug the effect of transfer learning explicitly for diverse medical tasks and with the small datasets, while only several hundred to thousand samples are available in many medical tasks. The study is somewhat limited to classification tasks only; however, it would be nice if the authors could provide insightful analysis for other medical tasks, such as segmentation, localization, quantification, and so on. Overall, I still feel that this is an incremental contribution to existing work with more fine-grained experiments based on the carefully chosen dataset. I think this is one of the main drawbacks of the study, which does not make sure about the claim posed in the paper. With such critical limitation, the study could fall short for NeurIPS; however, like other reviewers, I believe that the study is important and should be accessible by the greater community including NeurIPS. Considering all these aspects, I would like to stick to my previous score of 6 but tend to vote for accepting the paper.

Reviewer 2



Overall, I appreciated this paper very much. It addresses the problem of the use and misuse of transfer learning for medical applications and is therefore of high practical value. It is very clearly written and should be accessible to most attendants of NeurIPS. Some minor comments: - Page 2, line 38: microaneurysms appear as small red dots (not yellow). The yellow lesions (which may or may not be roundish) correspond to exudates. - Page 5, line 162: the authors certainly mean Figure 3 (even though Figure 8 can be seen as a confirmation). - Figures 2 and 3: What I found surprising is that the average CCA similarity across layers of the fully trained models is lower than the CCA similarity on the first layer before training, even for random initialization. Probably I missed something, but maybe the authors would like to comment on this, because others might find this intriguing as well. - Page 7, lines 197-199: I do not fully understand the second statement. I do not see from these two graphs that all larger systematically move less than smaller models. For instance, the different CBR variants show nearly the same value on the right plot for all three scenarios, and on the left plot, I also do not see a clear picture. ---- Update: I have read the answers by the authors and found this convincing. The paper is going to make a fine contribution to NeurIPS 2019.

Reviewer 3



I found this paper fascinating to read. I think this paper is quite strong, so my comments will be brief, aside from one recommendation (see Improvements) section that could potentially strengthen this work. Summary: This paper investigates the use of models pretrained on Imagenet that are used for transfer learning on medical applications. This is a ubiquitous practice, and this paper contains many (sometimes counter-intuitive) insights for the field of deep learning on medical images. Primarily, this work suggests that pretraining may be of little to no value for a sufficiently large (see improvements section for more) medical imaging dataset. Even more surprisingly, they find that relatively small models trained solely on the medical data are as good or better than large models like Resnet50 that are pretrained on Imagenet! I find this to be very surprising and exciting. The rest of the paper is devoted to some very creative explorations to explain this key finding. I think this paper should be read by everyone doing medical imagining as it contains numerous pieces of insight and should make us all reconsider what best practices ought to be. Originality Very original and creative paper. They underlying hypothesis (does transfer learning help in medical imaging) Clarity An extremely well written and lucid examination of transfer learning in medical imaging. The presentation flow very smoothly from section to section and is easy to follow. Significance Very high significance not only to medical imaging, but also to computer vision as a whole. Response to rebuttal: I have read the authors responses and found them suitable, and my score of 8 remains

[Author Response · NeurIPS 2019]

## Author Response to Reviews

Thank you for your time in reading the paper and the positive feedback! Below are responses for each reviewer.

### Response to Reviewer (id) 1

Thank you for your detailed reading of the paper and positive feedback!

*Similarity of representations from random init across random runs:* The greater variation in representational similarity for CBR-LargeT and CBR-LargeW likely arises due to the randomness of the initialization and the training process across different runs, which can affect similarity scores (see e.g. SVCCA, (Raghu et al, NeurIPS 2017)). It is therefore striking that for all the other models, the representations cluster so clearly into orange and blue in the plots from the paper – suggesting significant differences between representations learned from pretraining vs random initialization. Even for CBR-LargeT and CBR-LargeW, the mean similarity across the different runs is larger for the blue dots compared to the orange dots. With even more runs, we expect this distinction will be even clearer.

*Referenced paper:* Thank you for the reference to the paper on developing CNNs for tuberculosis screening and visualization, which we will add to the related work. We think it provides an interesting example of a non-standard architecture in the medical domain, but there are significant differences between it and our work. In particular, the paper states that 'the use of pretraining is outside the scope of the paper' – and does not compare the effect of pretrained weights vs random initialization. This is the core of the transfer learning question, and a central part of our paper. Additionally, the focus of the referenced paper is a specific medical application – tuberculosis – while our paper studies properties of pretraining and variation in architectures from representational viewpoints and across datasets, studying the effect of model overparametrization, the absence of traditional features such as Gabor filters, transfer in limited data regimes and convergence speed and pure scaling benefits of transfer.

### Response to Reviewer (id) 3

Thank you for the positive feedback, and for the Figure 3 reference typo comment (which is indeed what we meant) and microaneurysm color – we will edit these in the updated version!

*CCA similarity in Figures 2, 3:* The fact that CCA similarity in Figure 2 is lower than Figure 3 is likely due to two reasons. Firstly, multiple papers [1] Convergent Learning, (Li, Yosinski, Clune, Lipson, Hopcroft, ICLR 2016), [2] SVCCA, (Raghu et al, NeurIPS 2017), [3] Towards understanding learning representations (Wang, Hu, Gu, Wu, Hu, He, Hopcroft, NeurIPS 2018) have shown that deep networks trained on the same task are much less similar in the middle layers than at the start. This is even more the case when the networks are initialized differently (as suggested by Figure 2), and so when computing an average similarity over multiple layers, we reduce the overall CCA similarity score compared to just the first layer. Secondly, the CCA similarity score has some mild variation in scale across layers, due to different layers having different numbers of neurons (further discussed in [4] Insights on Representational Similarity, (Morcos, Raghu, Bengio, NeurIPS 2018), and [5] Similarity of Neural Network Representations Revisited (Kornblith, Norouzi, Lee, Hinton, ICML 2019). Therefore, a comparison of the CCA similarity scores across Figures 2, 3 might be harder to interpret directly (while the comparisons within Figures 2, 3 have a clear correspondence.)

*Page 7 lines 197-199:* Our intuition of larger networks moving less comes from Figure 3 and the filter visualizations in Figure 4. This phenomenon is consistent with related effects we have found since the submission in papers that we reference in the updated version of the paper: [6] Neural Tangent Kernel, (Jacot et al), [7] Are all layers created equal? (Zhang et al). In the submission, we computed the Euclidean distance between parameters at the start and end of training to further study this effect. However, Euclidean distance is sensitive to the total number of parameters and the resulting differences in initialization scalings, which we tried to address with a heuristic of dividing by the initialization norm $||w_0||$. But particularly in high dimensions, this remains a coarse measure of distance traveled, and while it was enough to show a difference between e.g. Resnet and the CBRs, it is not fine-grained enough to capture differences within the CBR family. In future work we hope to derive a more fine-grained distance metric.

### Response to Reviewer (id) 5

Thank you for the detailed review and positive feedback!

*Small Dataset Size:* We performed experiments on this important question with results shown in Table 3 in the Appendix of the submission. We simulated the small data regime by training our models on a small subset of the full training set but testing on the full test set. Our results highlight the following conclusions: for larger, overparametrized models e.g. Resnet50, there is a larger gap between transfer learning and random initialization for the smallest training dataset sizes (e.g. 5k training points). However, for smaller models, *both transfer and random initialization perform the same*, even for the 5k datapoints setting. This suggests that some of the gains from transfer learning in the small data regime are due to the overparametrization of the transfer-defined model architecture, rather than the reuse of pretrained features to prevent overfitting on the small dataset. In the updated version, we will add these conclusions to the main text.

[Meta-Review · NeurIPS 2019]

This work studied the merits of current transfer learning in medical imaging. It represents a strong empirical analysis of current state-of-the-art approaches, and leads to some somewhat surprising conclusions. Overall the reviewers agreed the work was strong and merited accepted.